# Analysis of anti-SARS-CoV-2 Omicron-neutralizing antibody titers in different vaccinated and unvaccinated convalescent plasma sources

The latest SARS-CoV-2 variant of concern Omicron, with its immune escape from therapeutic anti-Spike monoclonal antibodies and WA-1 vaccine-elicited sera, demonstrates the continued relevance of COVID-19 convalescent plasma (CCP) therapies. Lessons learnt from previous usage of CCP suggests focusing on early outpatients and immunocompromised recipients, with high neutralizing antibody titer units. Here, we systematically review Omicron-neutralizing plasma activity data, and report that approximately 47% (424/902) of CCP samples from unvaccinated pre-Omicron donors neutralizes Omicron BA.1 with a very low geometric mean of geometric mean titers for 50% neutralization $GM(GMT_{50})$ of ~13, representing a > 20-fold reduction from WA-1 neutralization. Non-convalescent subjects who had received two doses of mRNA vaccines had a $GM(GMT50)$ for Omicron BA.1 neutralization of ~27. However, plasma from vaccinees recovering from either previous pre-Omicron variants of concern infection, Omicron BA.1 infection, or third-dose uninfected vaccinees was nearly 100% neutralizing against Omicron BA.1, BA.2 and BA.4/5 with $GM(GMT_{50})$ all over 189, 10 times higher than pre-Omicron CCP. Fully vaccinated and post-BA.1 plasma (Vax-CCP) had a $GM(GMT_{50}) > 450$ for BA.4/5 and >1,500 for BA.1 and BA.2. These findings have implications for both CCP stocks collected in prior pandemic periods and for future plans to restart CCP collections. Thus, Vax-CCP provides an effective tool to combat ongoing variants that escape therapeutic monoclonal antibodies.

The SARS-CoV-2 Omicron variant of concern (VOC) (originally named VUI-21NOV-01 by Public Health England and belonging to GISAID clade GRA(B.1.1.529+BA.*) was first reported on 8 November 2021 in South Africa, and shortly thereafter was also detected all around the world. Omicron mutations impact 27% of T cell epitopes[1] and 31% of B cell epitopes of the Spike protein, while percentages for other VOC were much lower[2]. The Omicron variant has further evolved to several sublineages which are named by PANGO phylogeny using the BA alias: the BA.1 wave of Winter 2021-2022 has been suddenly replaced by BA.2

and BA.2.12.1 in Spring 2022, and by the BA.4 and BA.5 waves in Summer 2022.

The VOC Omicron is reducing the efficacy of all vaccines approved to date (unless 3 doses are delivered) and is initiating an unexpected boost in COVID-19 convalescent plasma (CCP) usage, with Omicron being treated as a shifted novel virus instead of a SARS-CoV-2 variant drift. Two years into the pandemics, we are back to the starting line for some therapeutic classes. Specifically, many Omicron sublineages escape viral neutralization by most monoclonal antibodies

✉ e-mail: daniele.focosi@gmail.com

(mAbs) authorized to date[3]. Despite the development of promising oral small-molecule antivirals (molnupiravir and nirmatrelvir), the logistical and economical hurdles for deploying these drugs worldwide have prevented their immediate and widespread availability, and concerns remain regarding both molnupiravir (both safety[4] and efficacy[5]) and nirmatrelvir (efficacy), especially in immunocompromised subjects. CCP was used as a frontline treatment from the very beginning of the pandemic. Efficacy outcomes have been mixed to date, with most failures explained by low dose, late usage, or both, but efficacy of high-titer CCP has been definitively proven in outpatients with mild disease stages[6,7]. Neutralizing antibody (nAb) efficacy against VOC remains a prerequisite to support CCP usage, which can now be collected from vaccinated convalescents, including donors recovered from breakthrough infections (so-called "hybrid plasma" or "Vax-CCP")[8]: pre-Omicron evidence suggest that those nAbs have higher titers and are more effective against VOCs than those from unvaccinated convalescents[9,10]. From a regulatory viewpoint, to date, plasma from vaccinees that have never been convalescent does not fall within the FDA emergency use authorization.

There are tens of different vaccine schedules theoretically possible according to EMA and FDA approvals, including a number of homologous or heterologous boosts, but the most commonly delivered schedules in the western hemisphere have been: (1) BNT162b2 or mRNA-1273 for 2 doses eventually followed by a homologous boost; (2) ChAdOx1 for 2 doses eventually followed by a BNT162b2 boost, and (3) Ad26.COV-2.S for 1 dose eventually followed by a BNT162b2 boost[11]. Many more inactivated vaccines have been in use in low-and-middle-income countries (LMIC), which are target regions for CCP therapy: this is feasible given the lower number of patients at risk for disease progression there (lower incidences of obesity, diabetes, and hypertension, and lower median age) and the already widespread occurrence of collection and transfusion facilities. Most blood donors there have already received the vaccine schedule before, after, or without having been infected, with a nAb titer generally declining over months[12]. Hence identifying the settings where the nAb titer is highest will definitively increase the efficacy of CCP collections. Variations in nAb titers against a given SARS-CoV-2 strain are usually reported as fold-changes in geometric mean titer of antibodies neutralizing 50% of cytopathic effect ($GMT_{50}$) compared to wild-type strains: nevertheless, fold-changes for groups that include non-responders can lead to highly artificial results and possibly over-interpretation. Rigorous studies have hence reported the percentage of responders as the primary

outcome and provided fold-changes of $GMT_{50}$ where the calculation is reasonable (100% responders in both arms)[13].

To date, the most rigorous data repository for SARS-CoV-2 sensitivity to antivirals is the Stanford University Coronavirus Antiviral & Resistance Database, but as of 24 July 2022 the tables there summarizing "Convalescent plasma" and "Vaccinee plasma" (https://covdb. stanford.edu/search-drdb) do not dissect the different heterologous or homologous vaccination schemes, the simultaneous occurrence of vaccination and convalescence, or the time from infection/vaccine to neutralization assay. Consequently, a more in-depth analysis is needed to better stratify CCP types.

In this work, we show that, in contrast to pre-Omicron CCP, plasma from Omicron convalescents who have been vaccinated is 100% neutralizing against Omicron: nAb titers are much higher in Vax-CCP than in CCP. These findings have implications for future collections, given the recent evidence supporting CCP efficacy in immunocompromised COVID-19 patients.

## Results

Our literature search identified 31 studies dealing with the original Omicron lineage (BA.1), which were then manually mined for relevant details: the PRISMA flowchart for study selection is provided in Fig. 1. Given the urgency to assess efficacy against the upcoming VOC Omicron, many studies (with a few exceptions[14–17]) relied on Omicron BA.1 pseudovirus neutralization assays, which, as opposed to living authentic virus neutralization assays, are scalable, do not require BSL-3 facilities, and provide results in less than 1 week. Plasma dilutions were expressed in the studies as $GMT_{50}$ of nAb, and fold reduction in $GMT_{50}$ compared to wild-type SARS-CoV-2 (e.g., WA-1) was the most common way of reporting changes, which reduces variability due to differences in the neutralization assays used. In comparing the large number of diverse studies with more than 100-fold plasma dilutional titers, we took the geometric mean of the individual study $GMT_{50}$, deriving a geomean of $GMT_{50}$ ($GM(GMT_{50})$).

Figure 2 and Table 1 summarize that neutralizing activity to WA-1 from CCP collected from subjects infected with pre-Alpha SARS-CoV-2 (Supplementary Table 1), Alpha VOC (Supplementary Table 2), Beta VOC (Supplementary Table 3), Delta VOC (Supplementary Table 4) or plasma from non-convalescent subjects vaccinated with 2 mRNA vaccine doses (Supplementary Tables 5 and 6). The same plasma types computed a geometric mean of multiple $GMT_{50}$ from many studies with about a 20-fold reduction against BA.1 geomeans compared to wild-type SARS-CoV-2 geomeans. CCP from uninfected vaccinees

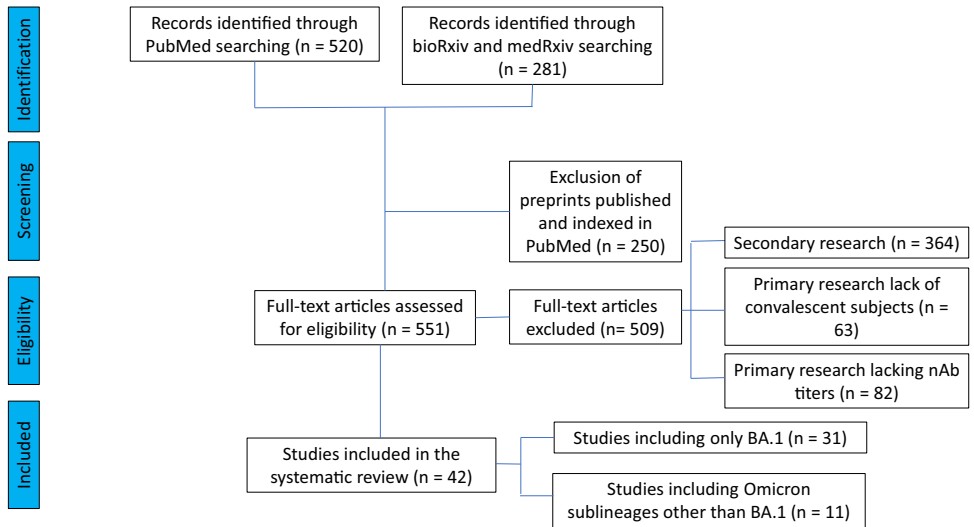

**Fig. 1 | PRISMA flowchart for the current study.** Number of records identified from various sources, excluded by manual screening, found eligible, and included according to subgroup analyses.

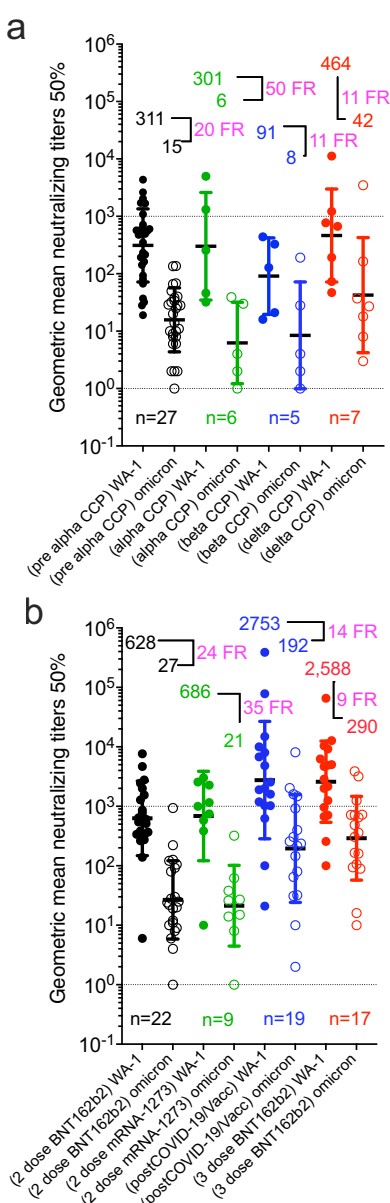

**Fig. 2 | Geometric mean neutralizing titers (GMT$_{50}$) against WA-1 versus Omicron BA.1 by study. a** unvaccinated convalescent plasma and **b** vaccinated plasma with or without COVID-19. Geomeans for entire study groups with neutralization of WA-1 in filled circles and of Omicron BA.1 in empty circles with geomeans (geometric standard deviation for error bars), fold reduction (FR) above data, and number of studies above x axis. All geomeans are not statistically significant in difference by multiple comparisons in Tukey's test. Source data are provided as a Source Data file.

receiving a third vaccine dose registered a geomean of the GMT$_{50}$ of 2,588 (or 10- fold higher nAb geomean of the GMT$_{50}$) to pre-alpha CCP viral assays. In this group the nAb geomean of the GMT$_{50}$ fold reduction against BA.1 was 9, but importantly the geomean of the GMT$_{50}$ was close to 300, similar to WA-1 inhibition by WA-1 CCP. The approximately 21-fold reduction in geomean of the GMT$_{50}$ from wild-type to BA.1 was reversed by the 10–15-fold increase in nAb geomean of the GMT$_{50}$ from either boosted (third-dose) vaccination or Vax-CCP.

In addition to the nAb GMT$_{50}$ levels showing potency, the percentage of individuals within a study cohort positive for any level of BA.1 neutralization shows the likelihood of a possible donation having anti-BA.1 activity. All studies but one tested a limited number of 20 to 40 individuals. The pre-Alpha CCP showed that most (18 of 27 studies)

had fewer than 50% of individuals tested within a study with measurable BA.1 neutralizing activity: only 2 out of 27 studies indicated that 100% of individuals tested showed BA.1 neutralization (Fig. 3). Likewise, most of the studies investigating Alpha and Beta CCP showed similar percent with nAb. Delta CCP had 6 of 7 studies with more than 50% BA.1 neutralization. The plasma from studies of the 2-dose mRNA vaccines indicated a more uniform distributive increase in percent of individual patients with measurable Omicron BA.1 nAb's. The stark contrast is pre-Omicron Vax-CCP, where 14 of 17 studies had 100% of individuals tested with anti-BA.1 nAb. The 3-dose vaccinee studies similarly had 12 of 17 studies with 100% measurable nAb.

Five studies directly compared anti-WA-1 to BA.1 nAb titers in nonvaccinated pre-Alpha, Alpha, Beta, and Delta CCP, and vaccinated plasma with the same nAb assay (Fig. 4). nAb GMT$_{50}$ against WA-1 was higher for Alpha CCP but lower for Beta CCP. nAb geomean of the GMT($_{50}$) against BA.1 was actually highest for delta CCP with geomean levels of 6, 6, 10 for pre-Alpha, Alpha, and Delta (Fig. 4, panel A). In these five studies, nAb geomean of the GMT($_{50}$) rose from 2-dose vaccinations to Vax-CCP to the 3-dose boosted vaccination. Importantly, for nAb, geomean of the GMT($_{50}$) against BA.1 were 14 to 103 to 195, respectively (Fig. 4b).

Another set of 9 matched vaccination studies inclusive of plasma collected after 2- and 3-dose schedules, as well as Vax-CCP, depicted a 20-fold rise in the geomean of the GMT($_{50}$) of anti-BA.1 nAb from the 2-dose vaccine to post COVID-19 vaccinees, and a 21-fold increase after the third vaccine dose. The pattern was similar for nAb geomean of the GMT($_{50}$) against WA-1 (Fig. 4c).

The AZD1222, 3-dose mRNA-1273, and Ad26.COV-2 vaccines were understudied, with 3 or fewer independent studies at different time points, reported in supplement Table 10. The GMT$_{50}$ nAb to BA.1 after 3- mRNA-1273 doses ranged 60 to 2000, with a 5-to-15-fold reduction compared with WA-1. GMT$_{50}$ of anti-BA.1 nAbs after AZD1222 vaccine was modest (-10 to 20), as with Ad26.COV-2 vaccine (-20 to 40). Two studies reported on post-COVID-19/post-mRNA-1273 with nAb GMT$_{50}$ against BA.1 of 38 and 272. Studies with 100% of individual patient samples neutralizing BA.1 included 2 3-dose mRNA-1273 studies, one AZD1222 study, and one post-COVID-19/post-mRNA-1273 study.

Few data exist for comparisons among different vaccine boosts. For CoronaVac® (SinoVac), three doses led to 5.1-fold reduction in anti-BA.1 nAb GMT$_{50}$ compared to wild-type[18], while for Sputnik V nAb titer moved from a 12-fold reduction at 6–12 months up to a sevenfold reduction at 2–3 months after a boost with Sputnik Light[19,20]. These in vitro findings have been largely confirmed in vivo, where prior heterologous SARS-CoV-2 infection, with and without mRNA vaccination, protects against BA.1 re-infection[21].

Eleven studies analyzed the efficacy of CCP and Vax-CCP against Omicron sublineages other than BA.1, i.e. BA.2 and BA.4/5 (summarized in Table 2, Fig. 5 and supplement Table 10–12). Pre-Omicron (mainly Delta) CCP neutralized less than 40% of BA.1, BA.2, and BA.4/5. Unvaccinated individuals in 3 studies only with BA.1 primary infections poorly neutralized WA-1, with about 75% neutralizations of BA.1, BA.2, and BA.4/5. The 3 vaccine doses of BNT162b2 in 8 studies had a 7-, 6- and 16-fold reduction from WA-1 GM(GMT$_{50}$) of 3,247 with percent of neutralizations all over 95%. Individuals from 11 populations with a BA.1 Vax-CCP had a 2-, 2-, and 8-fold reduction for WA-1 GM(GMT$_{50}$) of 3578, with 99% Omicron neutralizations. Importantly, GM(GMT$_{50}$) of BA.1 Vax-CCP was >1.5 times higher than that of pre-alpha CCP for WA-1. Omicron BA.1, BA.2, and BA.4/5 percent neutralization was over 99% for each. In conclusion, BA.1 Vax- CCP was both high-titer and high neutralization percent.

These studies largely confirm that Omicron CCP per se is poorly effective against the cognate or other Omicron sublineages[22] (with the lone exception of cross-reactions among lineages sharing L452 mutations[23] and broad-spectrum nAbs elicited by BA.5[24]). By contrast, both the homologous and the heterologous efficacy of Omicron

**Table 1 | Comparison of WA-1 to Omicron BA.1 nAb and percent with any Omicron BA.1 nAb amongst VOC CCP and vaccination status**

| Plasma type | Number of studies | WA-1 nAb $GMT_{50}$ | Omicron BA1 nAb $GMT_{50}$ | Fold reduction in nAb $GMT_{50}$ vs. Omicron BA.1 | Total number individuals in all studies | Total Omicron BA.1 neutralizing number | Omicron BA.1 neutralizing percent |
|---|---|---|---|---|---|---|---|
| pre-Alpha | 27 | 311 | 15 | 20 | 707 | 315 | 45 |
| Alpha | 6 | 301 | 6 | 50 | 64 | 21 | 33 |
| Beta | 5 | 91 | 8 | 11 | 37 | 19 | 51 |
| Delta | 7 | 464 | 42 | 11 | 94 | 69 | 73 |
| 2-dose BNT162b2 plasma | 22 | 628 | 27 | 24 | 432 | 202 | 47 |
| 2-dose mRNA-1273 plasma | 9 | 686 | 20 | 35 | 134 | 81 | 60 |
| Post-COVID-19/ full vacc plasma | 19 | 2753 | 192 | 14 | 279 | 243 | 87 |
| 3-dose BNT162b2 plasma | 17 | 2588 | 290 | 9 | 310 | 286 | 92 |

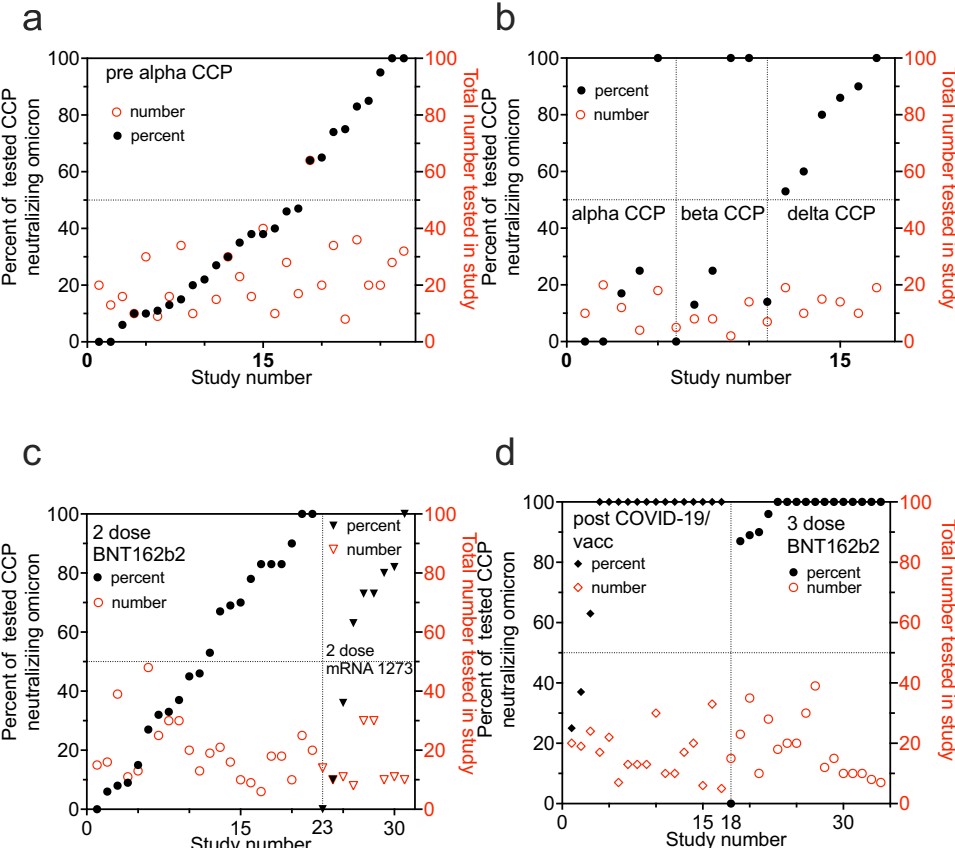

**Fig. 3 | Percent of individual plasma samples in each study showing any titer of Omicron BA.1 neutralization.** The percent of total samples within a study which neutralized Omicron BA.1 graphed in increasing percentages on left *y* axis with the total number of samples tested on the right *y* axis. **a** pre-Alpha CCP neutralization of Omicron BA.1; **b** Alpha, Beta and Delta CCP neutralization of Omicron BA.1; **c** 2-dose mRNA vaccines neutralization of Omicron BA.1 **d** post-COVID-19/post-vaccine (Vax-CCP) and uninfected 3-dose vaccine neutralization of Omicron BA.1. Source data are provided as a Source Data file.

Vax-CCP are universally preserved[15,25]. These findings have important implications if a Vax-CCP program is to be re-launched at the time of BA.2 and BA.4/5 waves. In particular, the emerging R346X-harboring BA.4.6, BA.4.7, and BA.5.9 sublineages show 1.5–1.9-fold reduction in $GMT_{50}$ by BA.1/2 Vax-CCP and 2.4–2.6 reduction by BA.5 Vax-CCP[26]. Of interest, Vax-CCP after 2 doses remains superior to 4-dose vaccine plasma in terms of nAb titers, and Vax-CCP with 3 vaccine doses is not consistently superior to Vax-CCP after 2 vaccines doses[27].

Differences exist among neutralization assay protocols used across studies. These differences are placed in context once fold reductions and percent neutralizations are used as reporting measures. The relative geometric mean titer with minimum and maximum titer levels is an alternative perspective. The live virus assays can be sorted from the pseudovirus assays along with the minimum and maximum extracted from the graphs or reported in data from the manuscripts. In general, ~ 80% of the live virus compared to

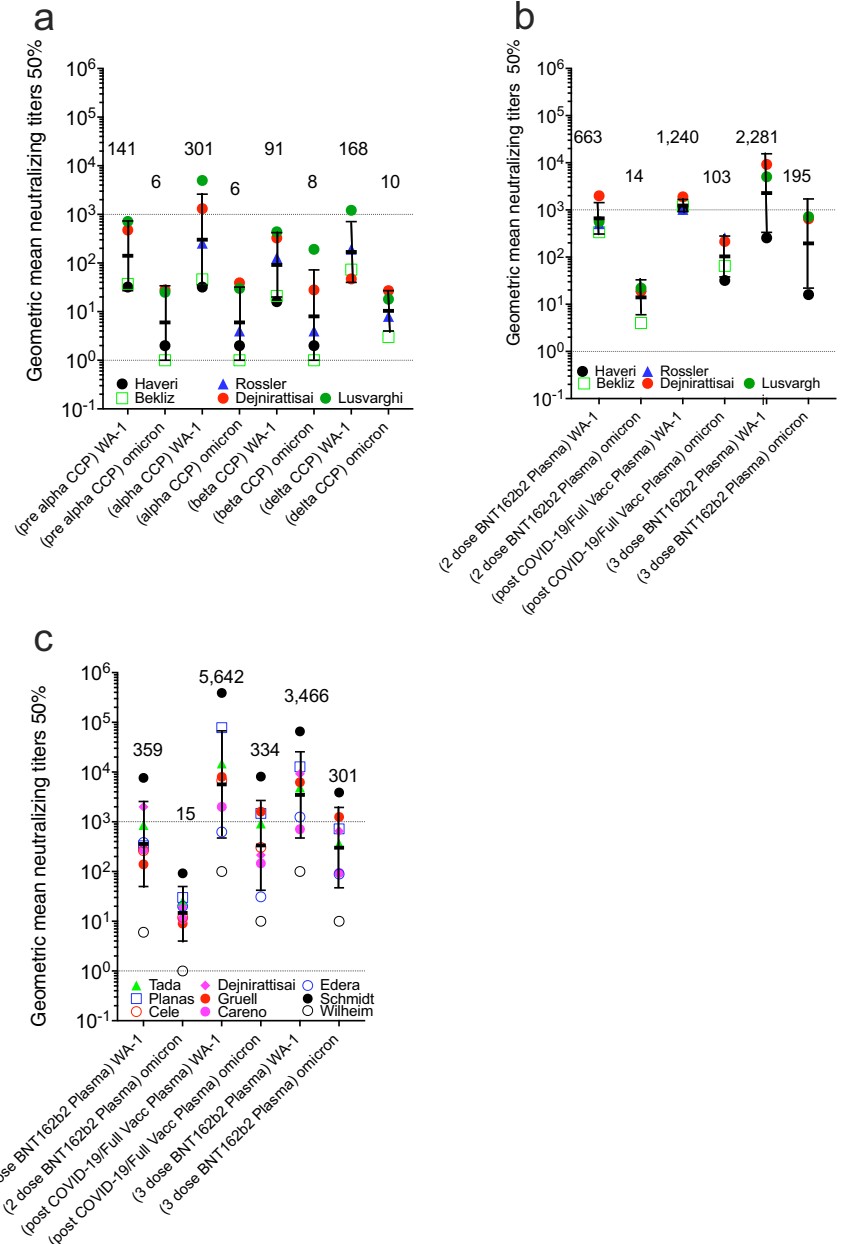

**Fig. 4 | Geometric mean neutralizing titers (GMT$_{50}$) of anti-WA.1 or anti-Omicron BA.1 neutralizing antibodies in plasma samples from 5 studies investigating diverse SARS-CoV-2 infecting lineage or vaccination status.** Five studies characterized **a** pre-Alpha, Alpha, Beta, and Delta CCP for Omicron BA.1 nAb compared to WA-1, and also **b** 2 or 3 doses BNT162b plasma, as well as post-COVID-19 plus BNT162b vaccine (Vax-CCP). **c** Nine additional studies looked at the same vaccine conditions in the first 5 comparing WA-1 nAb to Omicron BA.1 nAb (geometric standard deviation for error bars). Haveri et al. missing 2-dose BNT162b, Bekiliz et al and Rossler et al missing 3-dose BNT162b, Lusvarghi et al. missing post COVID-19/post full vaccine. Source data are provided as a Source Data file.

pseudovirus overlap in dilutional titer magnitude with the pseudovirus assays outlier measurement consistently higher in range from a few studies (Figs. 6–8). However, in some cases the minimum reported is not the limit of detection or quantification, but the lowest number in a group of SARS-CoV-2 antibody-positive participants. Graphical depiction of the minimum and maximum dilutional titers are shown in Supplementary Figs. 1–11.

## Discussion

Since nAbs are by definition antiviral, CCP with a high nAb GMT$_{50}$ is preferable, and there is now strong clinical evidence that nAb titers correlate with clinical benefit in randomized clinical trials[6,7]. Although nAb titers correlate with vaccine efficacy[28,29], it is important to keep in mind that SARS-CoV-2-binding non-neutralizing antibodies in CCP can

similarly provide protection via Fc-mediated functions[30,31]. However, such functions are harder to measure and no automated assay exists for use in clinical laboratories. Hence, whereas the presence of a high nAb GMT$_{50}$ in CCP is evidence for antibody effectiveness in vitro, the absence of nAb titer does not imply lack of protection in vivo where Fc effects mediate protection by other mechanisms such as antibody-dependent cell-mediated cytotoxicity, complement activation and phagocytosis.

The mechanism by which CCP from vaccinated COVID-19 convalescent individuals better neutralizes Omicron lineages is probably a combination of higher amounts of nAb and broader antibody specificity. Higher amounts of antibody could neutralize antigenically different variants through the law of mass action[32], whereby even lower affinity antibodies elicited to earlier variants

**Table 2 | Efficacy of CCP, vaccinee plasma, and Vax-CCP expressed as $GMT_{50}$ against Omicron sublineages**

| Plasma type | Number of studies | WA-1 nAb $GMT_{50}$ | Omicron BA.1 nAb $GMT_{50}$ (fold reduction from WA-1) | Omicron BA.2 nAb $GMT_{50}$ (fold reduction from WA-1) | Omicron BA.14/5 nAb $GMT_{50}$ (fold reduction from WA-1) | Total number individuals in all studies | Omicron BA.1; BA.2; BA.4/5 neutralizing percent |
|---|---|---|---|---|---|---|---|
| Pre-omicron CCP | 3 | 1338 | 133 (10 FR) | 132 (10 FR) | 177 (8 FR) | 50 | 35; 37;40 |
| BA.1 CCP | 3 | 71 | 366 (0.16 FR) | 180 (0.4 FR) | 82 (1 FR) | 74 | 66; 80; 76 |
| BA.4/5 CCP | 1 | 904 | 557 (2 FR) | 884 (1 FR) | 1047 (1 FR) | 13 | 100; 100; 100 |
| 3-dose BNT162b2 plasma | 8 | 3247 | 494 (7 FR) | 511 (6 FR) | 189 (16 FR) | 159 | 97; 98; 96 |
| Post-COVID-19-BA.1/full vacc plasma | 11 | 3578 | 1713 (2 FR) | 1830 (2 FR) | 454 (8 FR) | 142 | 99; 99; 99 |

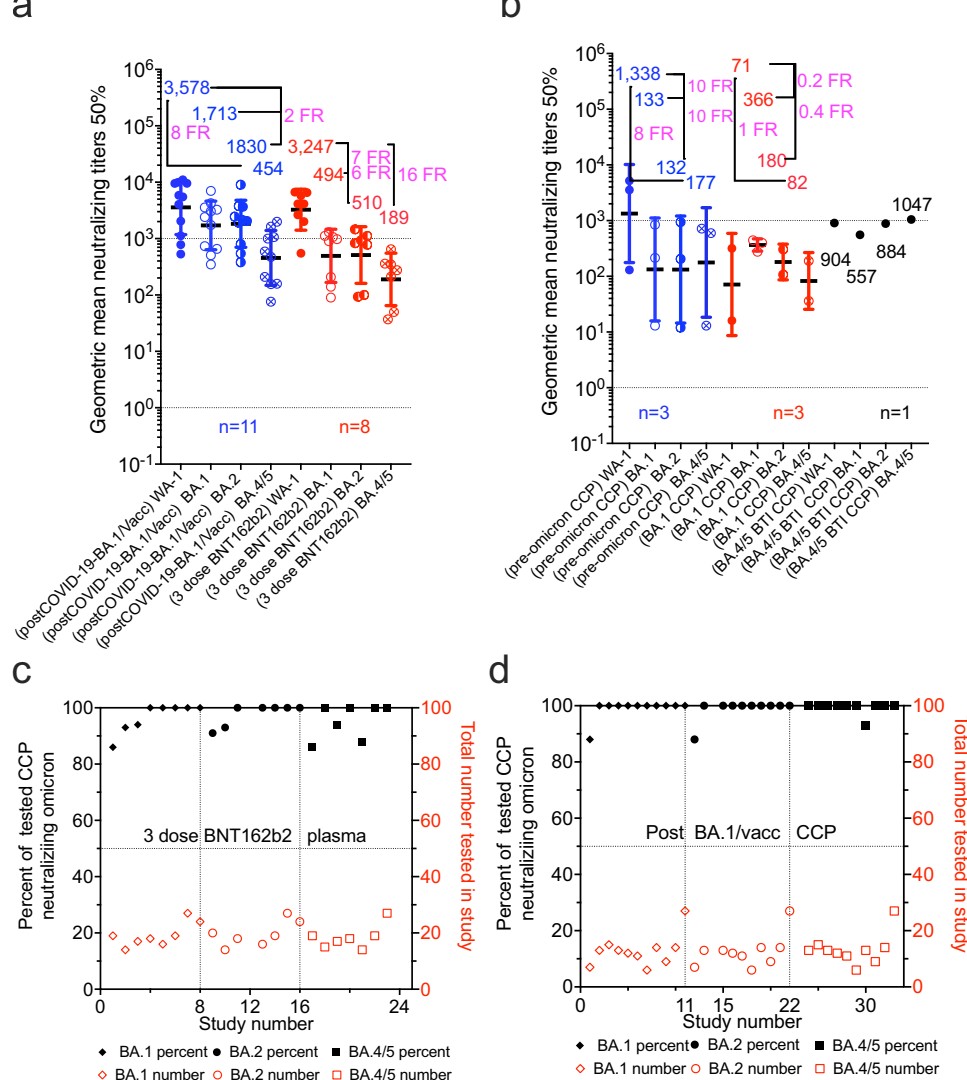

**Fig. 5 | Geometric mean neutralizing titers ($GMT_{50}$) against WA-1 versus Omicron BA.1, BA.2, or BA.4/5. a** boosted vaccinated plasma with or without BA.1 COVID-19 and **b** unvaccinated pre-Omicron or BA.1 CCP. Geomeans (geometric standard deviation for error bars), for entire study groups with neutralization of WA-1 in filled circles with Omicron in empty circles with geomeans and fold reduction (FR) above data and number of studies above x axis. All geomeans are not statistically significant in difference by multiple comparisons in Tukey's test. The percent of total samples within a study condition which neutralized Omicron graphed in increasing percentages on the left *y* axis with the number of total samples tested on the right *y* axis. **c** uninfected 3-dose vaccine neutralization of Omicron BA.1, BA.2, and BA.4/5; **d** post-COVID-19-BA.1/post-vaccine (Vax-CCP). Source data are provided as a Source Data file.

would bind to the Omicron variant as mass compensates for reduced binding strength to drive the reaction forward. In addition, vaccinated COVID-19 convalescent individuals would have experienced SARS-CoV-2 protein in two antigenically different forms: as part of intact infective virions generated in vivo during an infectious process and as antigens in vaccine preparations. As the immune system processes the same antigen in different forms, there are numerous opportunities for processing the protein in different

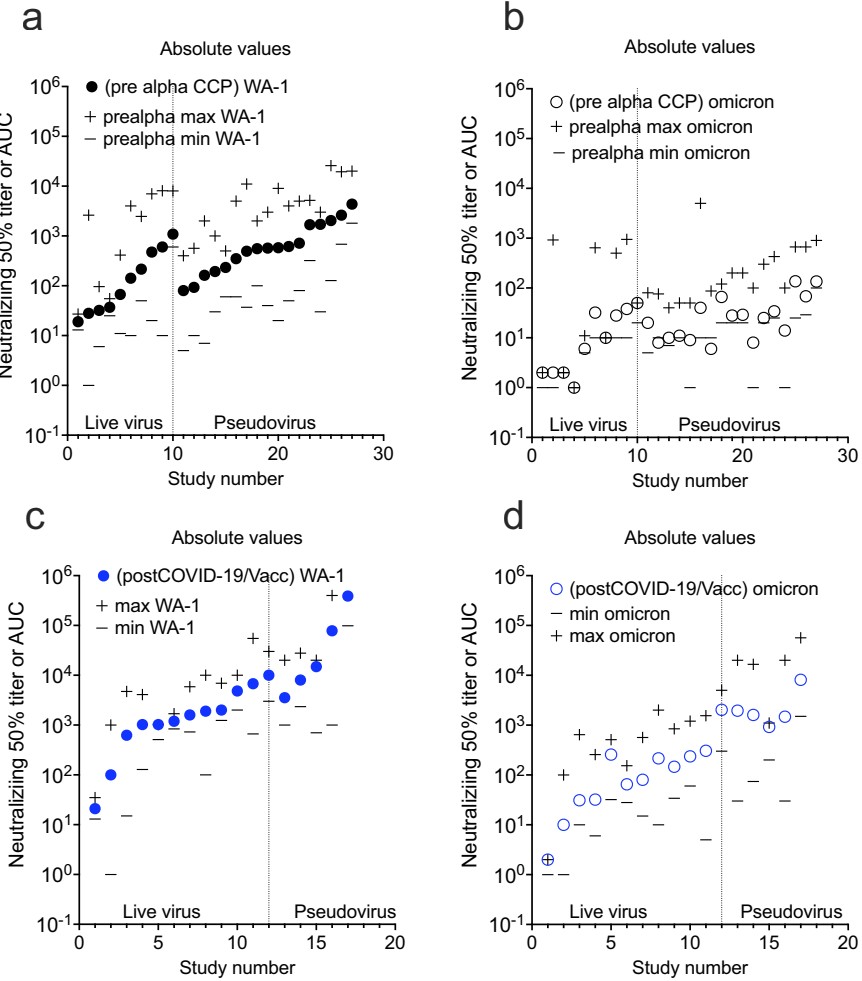

**Fig. 6 | GMT$_{50}$ from each study with minimum and maximum dilution titer also shown.** Live virus assays are on the left and pseudovirus assays on the right. **a** pre-alpha CCP against WA-1, **b** pre-alpha CCP against omicron BA.1, **c** post COVID-19/ Vax against WA-1. **d** post COVID-19/Vax against Omicron BA.1. Source data are provided as a Source Data file.

ways that can diversify the specificity of the immune response and thus increase the likelihood of eliciting antibodies that react with variant proteins. Structurally, it has been shown that third-dose mRNA vaccination induces mostly class 1/2 antibodies encoded by IGHV1–58;IGHJ3-1 and IGHV1-69;IGHJ4-1 germlines, but not the IGHV2-5;IGHJ3-1 germline, i.e. broadly cross-reactive class 3 antibodies seen after infection[33].

Our analysis provides strong evidence that, unlike what has been observed in Syrian hamster models[34], CCP from unvaccinated donors is unlikely (less than 50%) to have any measurable Omicron neutralization. Although the nAb GMT$_{50}$ threshold for clinical utility remains poorly defined, it is noticeable that low BA.1 nAb GMT$_{50}$ was generally detected in CCP after infection from pre-Omicron VOCs.

However, despite the huge heterogeneity of vaccine schedules, CCP from vaccinated and COVID-19 convalescent individuals (Vax-CCP) consistently harbors high nAb titers against BA.1 and novel sublineages if collected up to 6 months since the last event (either vaccine dose or infection). These Omicron-neutralizing levels are comparable in dilutional titers to that of WA-1 CCP neutralizing WA-1, but their prevalence is much higher at this time, facilitating recruitment of suitable donors. Pre-Omicron CCP boosted with WA-1-type vaccines induces heterologous immunity that effectively neutralizes Omicron in the same assays which rule in or out therapeutic anti-Spike monoclonal antibodies. Consequently,

prescreening of Vax-CCP donors for nAb titers is not necessary, and qualification of Vax-CCP units remains advisable only within clinical trials. A more objective way to assess previous infection (convalescence) would be by measuring anti-nucleocapsid (N) antibodies, but unfortunately, these vanish quickly[35,36]. Previous symptomatic infection and vaccination can be established by collecting past medical history (PMH) during the donor selection visit, which is cheaper, faster, and more reliable than measuring rapidly declining anti-N antibodies. Although there is no formal evidence for this, it is likely that asymptomatic infection (leading to lower nAb levels in pre-Omicron studies) also leads to lower nAb levels after vaccination compared to symptomatic infection, given that disease severity correlates with nAb titer[37,38]: hence asymptomatically infected donors missed by investigating PMH are also less likely to be useful.

The same reasoning applies to uninfected vaccinees receiving third-dose boosts, but several authorities, including the FDA, do not currently allow collection from such donors for COVID-19 therapy on the basis that the convalescent polyclonal and poly-target response is a prerequisite for efficacy and superior to the polyclonal anti-Spike-only response induced by vaccines. This may be a false premise for recipients of inactivated whole-virus vaccines (e.g., BBIBP-CorV or VLA2001): for BBIBP-CorV, where efficacy against Omicron is largely reduced[18,20,39], but the impact of boost doses is still unreported at the time of writing. Table 1 and Table 2 clearly

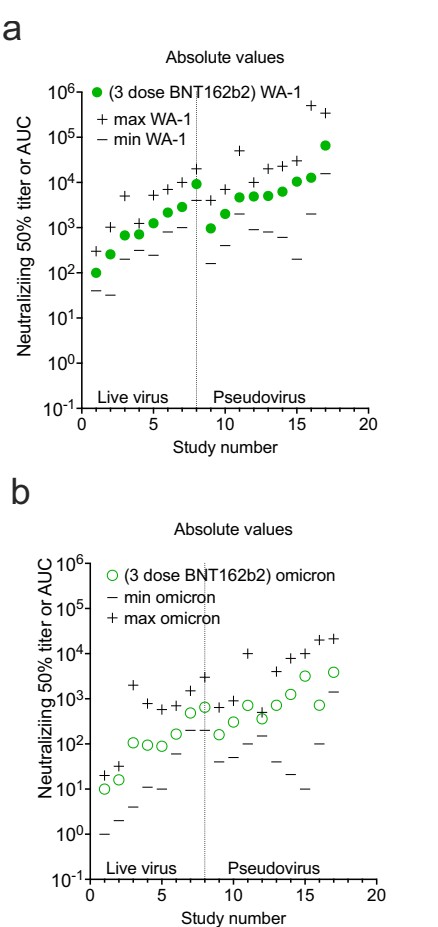

**Fig. 7 | GMT$_{50}$ from each study with minimum and maximum dilution titer also shown. a** 3-dose BNT162b2 against WA-1, **b** 3-dose BNT162b2 against Omicron BA.1. Source data are provided as a Source Data file.

show that 3 doses of BNT162b2 are enough to restore nAb levels against Omicron in the absence of SARS-CoV-2 infection.

Another point to consider is that information on nAb levels after the third vaccine dose has been almost exclusively investigated for only 1 month of follow-up, while studies on convalescents extend to more than 6 months. At present, it seems hence advisable to start from convalescent vaccinees rather than uninfected 3-dose vaccinees. This is also confirmed by immune escape reported in vivo after usage of vaccine (non-convalescent) plasma[40] despite very high nAb titers, likely due to restricted antigen specificity. Vaccine schedules with a delayed boost seem to elicit higher and broader nAb levels than the approved, short schedules[41–44], but this remain to be confirmed in larger series. The same is true for breakthrough infections from Alpha or Delta VOC in fully BNT162b2 vaccinated subjects[45], although variation in time from infection due to successive waves is a major confounder.

With the increase of Omicron seroprevalence in time, polyclonal intravenous immunoglobulins collected from regular donors could become a more standardized alternative to CCP, but their efficacy to date (at the peak of the vaccination campaign) is still 16-fold reduced against Omicron compared to wild-type SARS-CoV-2[46], and such preparations include only IgG and not IgM and IgA, which have powerful SARS-CoV-2 activity[47,48]. Nevertheless, the FDA recently reported efficacy of hyperimmune serum against BA.1, BA.2, BA.3, BA.2.12.1, and BA.4/5[49].

CCP collection from vaccinated convalescents (regardless of infecting sublineage, vaccine type, and number of doses) is likely to achieve high nAb titer against VOC Omicron, and, on the basis of lessons learnt with CCP usage during the first 2 years of the pandemic. Although in ideal situations one would prefer RCT evidence of efficacy against Omicron before deployment, there is concern that variants are generated so rapidly that by the time such trials commence this variant could be replaced by another. Given the success of CCP in 2 outpatient RCTs in reducing disease progression or hospitalization[6,7] and the loss of major mAb therapies for immunocompromised patients due to Omicron antigenic changes, the high titers in 2022 Omicron CCP collected from vaccinated convalescents provides an immediate option for COVID-19, especially in LMIC. Given the reduced hospitalization rate with Omicron compared to Delta[50], it is even more relevant to identify patient subsets at risk of progression in order to minimize the number needed to treat to prevent a single hospitalization. Using the same indications for use of mAb therapies while using the same (now unused) in-hospital facilities seems a logical approach.

## Methods

On August 11, 2022, we searched PubMed, medRxiv, and bioRxiv for research reports investigating the efficacy of CCP (either from vaccinated or unvaccinated donors) against SARS-CoV-2 VOC Omicron (pre)published after December 1, 2019, using English language as the only restriction. In PubMed, we used the search query "("convalescent plasma" or "convalescent serum") AND ("neutralization" or "neutralizing") AND "SARS-CoV-2"", while in bioRxiv and medRxiv we searched for abstract or title containing "convalescent, SARS-CoV-2, neutralization" (match all words). When a preprint was published, the latter was used for analysis. We also screened the reference lists of reviewed articles for additional studies not captured in our initial literature search. Articles underwent evaluation for inclusion by two assessors (D.F. and D.J.S.) and disagreements were resolved by a third senior assessor (A.C.). We excluded review articles, meta-analyses, studies reporting antibody levels by serological assays other than neutralization, as well as studies exclusively analyzing nAbs in vaccine-elicited plasma/serum from non-convalescent subjects. In unvaccinated subjects, convalescence was annotated according to infecting sublineage (pre-VOC Alpha, VOC Alpha, VOC Beta, VOC Delta, or VOC Omicron BA.1 sublineages). Given the heterologous immunity that develops after vaccination in convalescents, the infecting lineage was not annotated in vaccine recipients. In vaccinees, strata were created for 2 homologous doses, 3 homologous doses, or post-COVID-19 and post-vaccination (Vax-CCP). The type of viral assay (live or pseudovirus), time interval to a blood sample, geometric mean, minimum and maximum neutralizing 50% dilutional titer for WA-1 (pre-Alpha wild-type) and Omicron sublineages BA.1, BA.2 and BA.4/5, and number out of the total that neutralized Omicron was abstracted from study text, graphs, and tables. Prism v. 9.4 (GraphPad Software, San Diego, CA, USA) was used for data analysis.

Statistical significance between log$_{10}$ transformed geometric study means was investigated using Tukey's test. The multiple comparison test was a two-way ANOVA with alpha 0.05 on log-transformed GMT$_{50}$. The log-normal test was performed on"(pre alpha) WA-1", "(2-dose BNT162b2) omicron", "(alpha) WA-1", "(2-dose mRNA-1273) omicron", "(post-COVID-19/Vacc) WA-1", "(post-COVID-19/Vacc) omicron", "(3-dose BNT162b2) WA-1", and "(3-dose BNT162b2) omicron": these were normal by Anderson Darling, D'Agostino and Pearson, Shapiro-Wilk and Kolmogorov-Smirnov tests. These groups had too few results that a normality test was not able to be performed: "(alpha) omicron", "(beta) WA-1", "(beta) omicron", "(delta) WA-1", "(delta) omicron" and all of the plasma on WA-1 with BA.1, BA.2 and BA.4/5. Compiled data abstracted from the published studies is available in Supplementary Information in addition to the Source Data for figures.

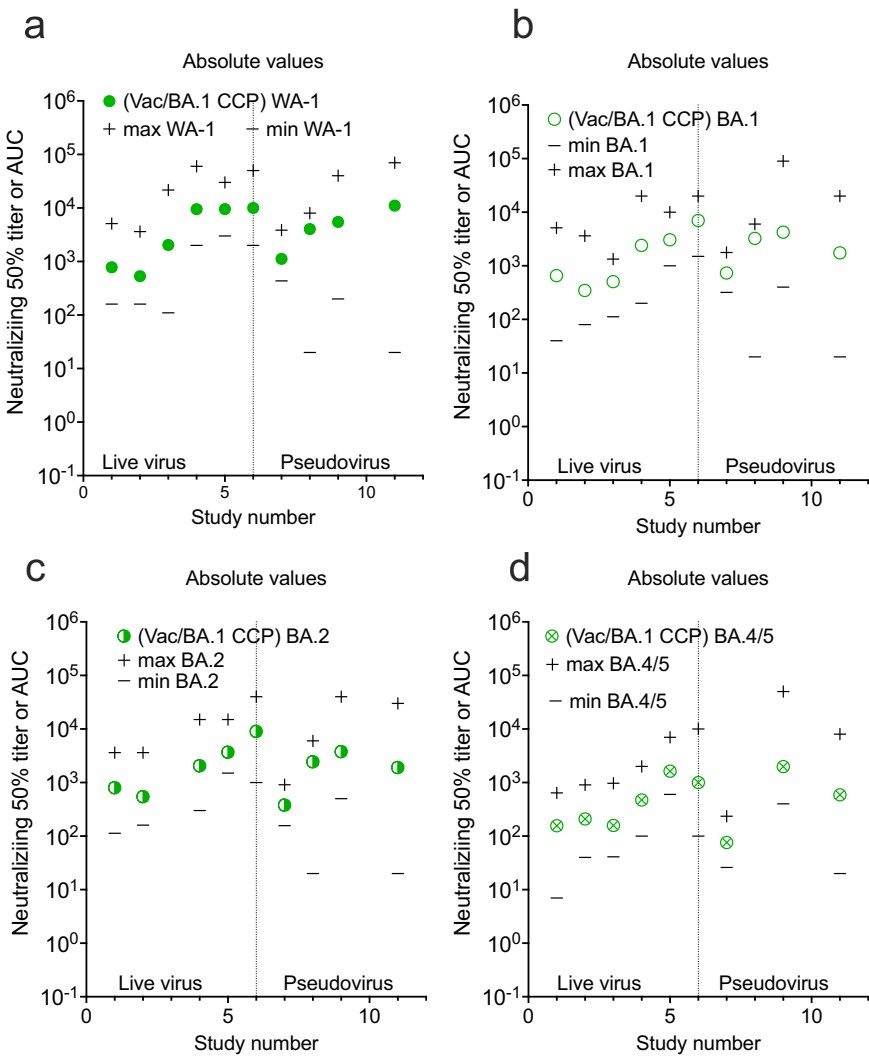

**Fig. 8 | GMT$_{50}$ from each study with minimum and maximum dilution titer also shown. a** post COVID-19-BA.1/Vac against WA-1, **b** post COVID-19-BA.1/Vac against Omicron BA.1, **c** post COVID-19 BA.1/Vac against Omicron BA.2, **d** post COVID-19 BA.1/Vac against BA.4/5. Source data are provided as a Source Data file.

### Reporting summary

Further information on research design is available in the Nature Research Reporting Summary linked to this article.

## Data availability

The raw numbers for charts and graphs are available in the Source Data file whenever possible. Source data are provided with this paper.

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

## Acknowledgements

The analysis was supported by the U.S. Department of Defense's Joint Program Executive Office for Chemical, Biological, Radiological and Nuclear Defense (JPEO-CBRND), in collaboration with the Defense Health Agency (DHA) (contract number: W911QY2090012) (D.S), with additional support from Bloomberg Philanthropies, State of Maryland, the National Institutes of Health (NIH) National Institute of Allergy and Infectious Diseases (NIAID) 3R01AI152078-01S1) (A.C).

## Author contributions

D.F. and M.J.J. conceived the manuscript; D.F., D.J.S., and M.F. analyzed the literature, curated tables, and wrote the manuscript; M.F. provided Fig. 1; D.J.S. provided Figs. 2–5. A.C. and M.J.J. revised the manuscript.

## Competing interests

The authors declare no competing interests.

## Additional information

**David J. Sullivan** [1], **Massimo Franchini** [2], **Michael J. Joyner** [3], **Arturo Casadevall** [1] & **Daniele Focosi** [4]

[1]Johns Hopkins Bloomberg School of Public Health and School of Medicine, Baltimore, MD 21218, USA. [2]Division of Transfusion Medicine, Carlo Poma Hospital, 46100 Mantua, Italy. [3]Department of Anesthesiology & Perioperative Medicine, Mayo Clinic, Rochester, MN 55902, USA. [4]North-Western Tuscany Blood Bank, Pisa University Hospital, 56124 Pisa, Italy. ✉e-mail: daniele.focosi@gmail.com

