## [Peer Review File · Nature Communications]

REVIEWER COMMENTS

Reviewer #1 (expertise in antibody responses and serological testing in SARS-COV2):

In this systematic review Focosi et al., review the current existing literature on Omicron neutralizing antibodies depending on vaccination status, as compared with infection. The authors propose the use of CCP as a more widespread therapy, which they supported by previous literature and clinical trials. In my view, the data that the authors present does not convincingly present this argument, for two main reasons: when they compare the nAb of convalescent vs vaccinees it appears that vaccinees (triple) have more Omicron nAb (Figure 4) than those present in convalescent individuals; and that the main CCP therapy now would be against Omicron subvariants based on a previous infection with another Omicron Subvariant (as they state at the end of their Results). I thought that the end of their Results section, focused on Omicron subvariants was perhaps the more relevant to the likely current scenarios that we will encounter from now onwards. I think that the authors only brush over these points and that these are very relevant now, where many countries are seeing a new wave of Omicron subvariants. I also understand that at the time of submission this was not the case, but I believe that this needs to be addressed. I am also missing more detail in their Methods and their inclusion/exclusion criteria.

More detailed points are below.

Abstract

Line 29: Omicron has been around since early December 2021 so I think 'novel' should not be in the abstract (line 29. I suggest 'latest'.

Line 29: do the authors mean immune escape, not antigenic escape?

Lines 41 and 42: Plasma from either boosted vaccinees or vaccination after pre-Omicron COVID-19 reads very complicated, please rephrase.

At the end of the Abstract, the use of CCP may be effective, but only under certain conditions, please state this clearly, as the last phrase reads as if any CCP would be useful when it is not the case (explained also by the authors). General statements like this when there is a lot of information on COVID-19 therapies need to be phrased carefully.

Introduction

Please double check the nomenclature of Omicron as in GISAID it appears as GRA(B.1.1.529+BA.*) which will include all sub-variants. If the authors are doing a metanalysis of the original Omicron then please clearly say so.

Line 49 suggests that Omicron appeared first in South Africa and 'spread all around the world' while it is plausible that it was circulating in several countries, and SA simply reported it first. Please rephrase.

Lines 49-50: the authors cite pre-prints (Ref 2) and thus they should rephrase their statements 'predicted to' or 'proposed to' or similar. In Line 50 please rephrase the use of 'significantly' if it does not reflect statistical findings.

Lines 76-78 is there anything missing the phrase? Also, what is the infrastructure of transfusions in LMIC, how much would this need to be expanded for CCP therapy to have an effect at the target population level?

Lines 78-80 please add a reference for this statement, this is quite a crucial and interesting piece of information.

Until I read the last paragraph of Results I was missing the clarification by the authors with regards to which Omicron were they referring to (original Omicron or any of its subvariants). For example, Sotrovimab was authorised in December 2021 and in April 2022 it stopped being authorised due to the dominance of BA.2 in the US at the time, for example. As the authors state in their last phrase (lines 187-188), the efficacy of CCP from an Omicron sub-variant (BA.2 at the

time of submission) on other sub-variants will be the most common scenario.

Methods

Did the authors test for the normality of the data to choose a Tukey test?

Results

Please have a look at Seow et al., 2022 (doi: 10.1016/j.celrep.2022.110757).

Please explain the exclusion criteria used in Figure 1 after the screening of 55 papers. Why were those first 21 excluded, then the n=5 full-text articles excluded, and then 29 of 34 included?

Please clarify if the values given are ID50 and define all acronyms when first mentioned (e.g. FR).

I kindly suggest that a legend is added regarding the empty vs filled circles in Figures 2.

In line 135 there's an extra 'had'. Figure 3D shows that several studies had 100 patients with 100% CCP neutralizing Omicron and in Figure 3C two studies. Are these numbers correct?

Figure 4B is missing the measure of Lusvarghi in the second and 5th groups – or if it is there it is not obvious to the reader. Can the authors clarify?

I suggest rearranging Figure 4 as per the previous Figure 2 i.e. the effects on WA-1 and on Omicron next to each other per condition. This Figure also suggests that neutralizing antibody titres from CCP (A) are lower than those that come from vaccination (B).

Line 166 please check the phrasing.

Discussion

I suggest that the authors refer to their results Figures when discussing the data. Line 198 please define ADCC. Please also provide a reference for the law of mass action.

With regards to symptomatic infection and nAb levels – the evidence that the authors refer to are all pre-Omicron (one reference is 2020). Omicron has very different symptoms to those of previous variants, and the correlation of symptoms to nAbs may be different too- I believe they need to clarify this point.

Please add references to IgM and IgA being powerful against SARS-CoV-2 (lines 252-253).

Reference 52 is an equine study and I cannot see where they mention Omicron having less hospitalization rates than Delta– this reference may be relevant in other context, but I cannot see how it is here, please place appropriately.

I am missing in the Discussion how the authors propose the use of CCP in LMIC as they state in their Introduction in lines 76-78. CCP can indeed be useful, but the data they present suggest that vaccination + booster has higher nAb against Omicron.

Reviewer #2 (expertyise in antibody responses, vaccines, Covid convalescent plasma):

Focosi et al analyze in their study the potential utility of convalescent plasma in the treatment of Omicron infection based on a literature research of published records of Omicron neutralization in CCP. While the topic is timely, the overall approach interesting, and the basic message informative for the field, the study lacks in methodological detail and needs major re-writing to qualify.

1. The methodology is not clear as described. How was the search for the papers conducted? Was this a systematic search? Which keywords were used? The authors show in Figure 1 a Flowchart of the inclusion/exclusion process but give no information on the criteria. The entire process as detailed in Figure 1 needs to be explained in the methods section.

2. Utilizing the mean neutralization titer of each study is a straightforward approach but may create some bias depending on the sample sizes of each study. The authors need to lay out how samples sizes and variability within studies are dealt with.

3. The studies differ in the type of neutralization assay used and therewith absolute titers may

vary substantially. The authors should consider an approach where within each study relative titers based on minimum and maximum titers observed are created. As is, the authors refer interchangeably to IC50 and GMT dependent to which study they refer. This makes the overall analysis hard/impossible to follow. FR for fold reduction is an unusual abbreviation, spelling the term out would increase readability.

4. Line 106: The mean neutralizing titer for WA-1 (pre-Alpha wild-type), Omicron and number out of total that neutralized Omicron was abstracted from studies. Figure 4 illustrates the need of harmonizing the neutralization titer readout other than using the mean of each study. Relative titers as suggested above may be a solution. An amended Figure 4 should also address dynamic ranges of the observed titer and comment on within study variability.

5. The opportunity to go beyond a simple neutralization titer assessment was missed. Considering all the data the authors extracted from the published studies, a meta-analysis to assess factors associated with neutralizing titers against omicron / wt could be considered and may provide interesting insights.

6. The analysis is kept mostly descriptive, more quantitative measures would be important to incorporate. In fact, the interpretation of the data given in the results is not on all occasions the one an independent reader may see when looking at the figures ...

7. The manuscript suffers from confusing wording and many comments that are unclear, incorrect, misleading, or not supported by references. I strongly recommend drastic editing of the manuscript. Here some examples of such comments:

a. Line 29: "The novel SARS-CoV-2 Omicron, with its antigenic escape from unboosted vaccines and therapeutic monoclonal antibodies, demonstrates the continued relevance of COVID19 convalescent plasma (CCP) therapies" Given the current information on clinical use of CCP it is very bold to state "demonstrates the continued relevance of COVID19 convalescent plasma (CCP) therapies". This needs to be reworded.

b. "Line 43 Thus, CCP provides an effective tool to combat the emergence of variants that defeat therapeutic monoclonal antibodies". Why/how would CCP combat the emergence of variants? This statement needs explanation and is not suited for the abstract.

c. "Line 96 Consequently, a more in-depth analysis is needed to better stratify the populations." This statement is unclear and suggests that the present study provides a means of clinically relevant stratification of CCP, which is not the case.

8. Tables 2 through 9 are not mentioned in the results!

9. Line 113 "...with a few exceptions". Please clarify: which studies used replication competent virus?

10. Line 118 "...nAb titers of 850 to 2,000" Please clarify: NT50? GMT? Nomenclature keeps changing in the result section. Please consider relative titers for comparability.

11. Line 119 "The same plasma" Please clarify: Does this refer to the Beta VoC infected CCP?

12. Line 154 onwards: "Few data exist for comparisons among different vaccine boosts. For CoronaVac® (SinoVac), three doses led to 5.1 FR in nAb titer 20, while for Sputnik V nAb titer moved from a 12-fold reduction at 6-12 months up to a 7-fold reduction at 2-3 months after a boost with Sputnik Light." Please clarify "5.1 FR in nAb titer 20", what does this refer to exactly?

13. Line 168: "IC50 values to mean 1:2929 at around 9-12 days, which were higher than the mean peak IC50 values of BNT162b2 vaccines" Referring here to IC50 makes these data difficult to other parts in the result section. As mentioned above, working with relative titers would resolve this issue.

14. Line 34 approximately 50% (426/841). Please clarify: are these treated patients?

15. Line 35 "...about 30". Please clarify: Does this mean 30 NT50? Or GMT?

16. Line 35/36 Two doses of mRNA vaccines had a similar 50% percent neutralization with more than doubling of Omicron neutralization mean titer (about 60) Please clarify: This refers to neutralizing activity of CCP of vaccinated patients? How long after vaccination? (Latter needs to be also stated in results).

17. Line 121: "The nAb FR against Omicron was now 10 to 20 ..." What do the authors trying to say?

18. Line 213 "...likely (less than 50%)". Please clarify, less than 50% or likely not?

19. Figure legend 2. Panel A and B are not explained.

20. Figure legend 4. Description of panel C is missing

21. Table 4-10: Please clarify in Table what "mean" stands for. Geometric mean neutralizing titer (GMT)? What does time stand for? Sampling time after last vaccination dose? Fold drop means fold

reduction to WA-1? This is not clearly labelled in all tables. Please use the same terminology in text and tables.

RESPONSE TO REVIEWERS' COMMENTS

Response to Reviewer #1 :

In this systematic review Focosi et al., review the current existing literature on Omicron neutralizing antibodies depending on vaccination status, as compared with infection. The authors propose the use of CCP as a more widespread therapy, which they supported by previous literature and clinical trials. In my view, the data that the authors present does not convincingly present this argument, for two main reasons: when they compare the nAb of convalescent vs vaccinees it appears that vaccinees (triple) have more Omicron nAb (Figure 4) than those present in convalescent individuals; and that the main CCP therapy now would be against Omicron subvariants based on a previous infection with another Omicron Subvariant (as they state at the end of their Results).

Please note that we are proposing Vax-CCP, not CCP, for treatment. We agree that plasma from triple vaccinees regularly contains higher anti-Spike nAb levels than plasma from convalescents only, but plasma from vaccinees only (non-convalescent) cannot be used under current regulations. As we have discussed in the text, the occurrence of both conditions (vaccination and convalescence) induces heterologous immunity, which is able to protect against variants other than the one that infected the convalescents. Our new analysis of 11 different studies of fully vaccinated individuals after BA.1 COVID-19 noted only a 2-fold omicron GM(GMT₅₀) reduction for BA.1 or BA.2 and 8-fold with BA.4/5, with GM(GMT₅₀) over 1500 for BA.1 and BA.2 and over 450 for BA.4/5. This contrasts to the WA-1 GM(GMT₅₀) of 311 with original WA-1 CCP and approximates the over 2000 GM(GMT₅₀) for WA-1 virus with triple shot BNT162b2 and Vax-CCP with pre-Omicron VOC.

I thought that the end of their Results section, focused on Omicron subvariants was perhaps the more relevant to the likely current scenarios that we will encounter from now onwards. I think that the authors only brush over these points and that these are very relevant now, where many countries are seeing a new wave of Omicron subvariants. I also understand that at the time of submission this was not the case, but I believe that this needs to be addressed.

We totally agree that presently, 3 months after initial submission, the pandemic has progressed, and Omicron should be the focus of the research. We have hence largely expanded the Omicron section of the manuscript evidences emerged in the meanwhile with BA.1, BA.2 and BA.4/5. We were able to extract data for the three omicron variants from 11 distinct manuscripts which form the basis for a new Table 2 on neutralization of the new omicron lineages.

I am also missing more detail in their Methods and their inclusion/exclusion criteria.

As also requested by Reviewers #2, we have clarified the literature selection criteria in the revised PRISMA flowchart. We have specified the exact search queries and the reasons for exclusions. Excluded studies often did not report numbers for GMT with reliance on fold reduction for their publication.

Line 29: Omicron has been around since early December 2021 so I think 'novel' should not be in the abstract (line 29. I suggest 'latest'.

Changed as requested.

Line 29: do the authors mean immune escape, not antigenic escape?

Yes, changed as suggested.

Lines 41 and 42: Plasma from either boosted vaccinees or vaccination after pre-Omicron COVID-19 reads very complicated, please rephrase.

Rephrased

At the end of the Abstract, the use of CCP may be effective, but only under certain conditions, please state this clearly, as the last phrase reads as if any CCP would be useful when it is not the case (explained also by the authors). General statements like this when there is a lot of information on COVID-19 therapies need to be phrased carefully.

We agree with the reviewer and have rephrased the statement. The bottom line is that high-titre plasma from vaccinated convalescents is effective against Omicron sublineages. It is likely useful until there is ongoing viral replication and the recipient remains seronegative.

Please double check the nomenclature of Omicron as in GISAID it appears as GRA(B.1.1.529+BA.) which will include all sub-variants. If the authors are doing a metanalysis of the original Omicron then please clearly say so.*

Changed as suggested, since the focus of our manuscript is the entire Omicron VOC. We are relying over PANGOLIN nomenclature (using the BA alias) to identify sublineages.

Line 49 suggests that Omicron appeared first in South Africa and 'spread all around the world' while it is plausible that it was circulating in several countries, and SA simply reported it first. Please rephrase.

We agree with the hypothesis, but believe that "reported" is different from "originated". The word is actually a recognition for the sequencing effort run in South Africa, not a stigma. We have anyway changed "spread" into "detected".

Lines 49-50: the authors cite pre-prints (Ref 2) and thus they should rephrase their statements 'predicted to' or 'proposed to' or similar.

The preprint at Ref 2 has now been fully published in a peer-reviewed journal, so we feel rephrasing is not necessary

In Line 50 please rephrase the use of 'significantly' if it does not reflect statistical findings.

Rephrased into "much".

Lines 76-78 is there anything missing the phrase?

No, but we hope to have now further clarified the phrase.

Also, what is the infrastructure of transfusions in LMIC, how much would this need to be expanded for CCP therapy to have an effect at the target population level?

We have now clarified in the text that collection and transfusion facilities are already well implemented in LMIC, and that the number of COVID19 patients at risk for disease progression (that would hence require CCP transfusion) is lower in LMIC than in developed economies (lower incidence of comorbidities and lower median age).

Lines 78-80 please add a reference for this statement, this is quite a crucial and interesting piece of information.

Reference added.

Until I read the last paragraph of Results I was missing the clarification by the authors with regards to which Omicron were they referring to (original Omicron or any of its subvariants). For example, Sotrovimab was authorised in December 2021 and in April 2022 it stopped being authorised due to the dominance of BA.2 in the US at the time, for example. As the authors state in their last phrase (lines 187-188), the efficacy of CCP from an Omicron sub-variant (BA.2 at the time of submission) on other sub-variants will be the most common scenario.

To account for evolution of the Omicron VOC in the meanwhile the study was designed, we replicated our study taking into account the main Omicron VOCs reported to date. Given the search output (a minority of studies focusing on Omicron sublineages), we decided to leave sublineage analysis separated from the rest of the Tables (which we have now clarified only refers to BA.1 lineage), gathering them in Table 2. We have clarified the scope of the review in title, abstract, introduction and discussion to differentiate BA.1, BA.2 and BA.4/5

Did the authors test for the normality of the data to choose a Tukey test?

The log normal test was performed on "(pre alpha) WA-1", "(2 dose BNT162b2) omicron", "(alpha) WA-1", "(2 dose mRNA-1273) omicron", "(postCOVID-19/Vacc) WA-1", "(postCOVID-19/Vacc) omicron", "(3 dose BNT162b2) WA-1", and "(3 dose BNT162b2) omicron": these

were normal by Anderson Darling, D'Agostino and Pearson, Shapiro-Wilk and Kolmogorov-Smirnov tests. These groups had too few results that a normality test was not able to be performed: "(alpha) omicron", "(beta) WA-1", "(beta) omicron", "(delta) WA-1", and "(delta) omicron"

Please have a look at Seow et al., 2022 (doi: 10.1016/j.celrep.2022.110757).

The Seow *et al* study was excluded during study selection because it exclusively deals with serum/plasma from vaccinated but non-convalescent subjects. As such, it falls outside the scope of this manuscript.

Please explain the exclusion criteria used in Figure 1 after the screening of 55 papers. Why were those first 21 excluded, then the n=5 full-text articles excluded, and then 29 of 34 included?

We have better detailed reasons for study exclusions within the revised PRISMA flowchart. The numbers are consistent and replicable. The output is split into BA.1 versus non-BA.1 Omicron

Please clarify if the values given are ID50 and define all acronyms when first mentioned (e.g. FR).

All values given in Figures and Tables are geometric mean titers (GMT) of 50% neutralizing antibodies (nAb) (GMT50). We have now introduced geometric mean of GMT50 (GM(GMT50)) whenever needed in place of arithmetic mean.

I kindly suggest that a legend is added regarding the empty vs filled circles in Figures 2.

The original legend should have included "WA-1 in filled circles with Omicron in empty circles". The legend has now been revised to reflect the new geomeans.

In line 135 there's an extra 'had'.

Removed.

Figure 3D shows that several studies had 100 patients with 100% CCP neutralizing Omicron and in Figure 3C two studies. Are these numbers correct?

The numbers are correct with the "3 dose vaccine" and "post COVID-19/vaccination" groups with high number of 100% neutralizing. See current Supplementary tables 7 and 8.

Figure 4B is missing the measure of Lusvarghi in the second and 5th groups – or if it is there it is not obvious to the reader. Can the authors clarify?

Such data were missing from the published paper.

	Haveri	Bekliz	Rosler	Dejnirattisai	Lusvarghi
(2 dose BNT162b2 Plasma) WA-1 mean		338	512	1993	562
(post COVID-19/Full Vacc Plasma) WA-1 mean	1024	1190	1000	1899	
(3 dose BNT162b2 Plasma) WA-1 mean	290			9219	5029
(2 dose BNT162b2 Plasma) omicron mean		4	16	19	22
(post COVID-19/Full Vacc Plasma) omicron mean	32	66	250	215	
(3 dose BNT162b2 Plasma) omicron mean	24			649	718

I suggest rearranging Figure 4 as per the previous Figure 2 i.e. the effects on WA-1 and on Omicron next to each other per condition. This Figure also suggests that neutralizing antibody titres from CCP (A) are lower than those that come from vaccination (B).

The values were rearranged so that omicron is next to WA-1. The superiority of vaccine plasma vs CCP is real, but the focus here is on the much greater superiority of VaxCCP (panel C) vs. the other two products.

Line 166 please check the phrasing.

Paragraph moved to Table 11.

I suggest that the authors refer to their results Figures when discussing the data.

We have now added references to Figures throughout the revised manuscript.

Line 198 please define ADCC.

Defined.

Please also provide a reference for the law of mass action.

Provided.

With regards to symptomatic infection and nAb levels – the evidence that the authors refer to are all pre-Omicron (one reference is 2020). Omicron has very different symptoms to those of previous variants, and the correlation of symptoms to nAbs may be different too- I believe they need to clarify this point.

Unfortunately, at the best of these authors' knowledge there is no study yet correlating severity of symptoms (which we agree is generally milder with the Omicron VOC than with former VOCs for multiple reasons) with post-infection neutralizing antibody GMTs, likely because vaccine coverage is very high these days and acts as a confounder. Hence, we have added a caveat that those assumptions are based on pre-Omicron studies.

Please add references to IgM and IgA being powerful against SARS-CoV-2 (lines 252-253).

Relevant references added.

Reference 52 is an equine study and I cannot see where they mention Omicron having less

hospitalization rates than Delta– this reference may be relevant in other context, but I cannot see how it is here, please place appropriately.

Apologies for the mistake. We have now deleted such misplaced reference.

I am missing in the Discussion how the authors propose the use of CCP in LMIC as they state in their Introduction in lines 76-78. CCP can indeed be useful, but the data they present suggest that vaccination + booster has higher nAb against Omicron.

As we have now stated in the Introduction, plasma from boosted subjects who have never been convalescents cannot be currently used therapeutically because of regulatory restrictions. Furthermore, it largely lacks antibody classes other than IgG. So VaxCCP remains the only feasible option when it comes to high nAb titer units. We have also now better clarified in Introduction the feasibility of CCP in LMIC (less patients at risk for disease progression, minimal upgrade of the collection and transfusion facilities)

Responses to Reviewer #2

1. The methodology is not clear as described. How was the search for the papers conducted? Was this a systematic search? Which keywords were used? The authors show in Figure 1 a Flowchart of the inclusion/exclusion process but give no information on the criteria. The entire process as detailed in Figure 1 needs to be explained in the methods section.

As now better clarified in title, abstract and Introduction, this was a systematic review. The exact query has now been added to the text. We have also better detailed the PRISMA flowchart to account for the exact exclusion reasons and stratified the output (BA.1 vs. non-BA.1 Omicron).

2. Utilizing the mean neutralization titer of each study is a straightforward approach but may create some bias depending on the sample sizes of each study. The authors need to lay out how samples sizes and variability within studies are dealt with.

We think that the last 3 columns on the right of each table showing the total number of individuals in a study along with the number showing omicron neutralizing activity as well as percent are an effective way to account for sample size in each study. While a weighted average can be performed, we have chosen to report the mean neutralization as a weighted analysis can also skew to large sampling studies which may not add precision on the geometric means. We have nevertheless now reported also the min-max GMT in a separate Supplementary Figures as well as the source data file.

3. The studies differ in the type of neutralization assay used and therewith absolute titers may vary substantially. The authors should consider an approach where within each study relative titers based on minimum and maximum titers observed are created. As is, the authors refer interchangeably to IC50 and GMT dependent to which study they refer. This makes the overall analysis hard/impossible to follow. FR for fold reduction is an unusual abbreviation, spelling the term out would increase readability.

All tables and figures exclusively report fold reductions in GMT for 50% neutralizations (GMT50 not IC50), making them consistent.

Albeit differences exist among neutralization assay protocols used across studies, we feel those differences are placed in context once fold-reductions and percent neutralizations are used as reporting measures. We do agree that a relative titer with minimum and maximum is an alternative. We have sorted the live virus assays from the pseudovirus assays along with the minimum and maximum extracted from the graphs or reported in data from the manuscripts. We included a few illustrative graphs of the absolute values for minimum and maximum. However, in some cases the minimum reported is not the limit of detection or quantification, but the lowest number in a group of SARS-CoV-2 antibody positive participants. In a newly created supplementary Table we graphed this transformation. We included a few representative graphs of WA-1 and omicron expressed in absolute values for the main paper.

FR has been spelled out at all instances as requested.

4. Line 106: The mean neutralizing titer for WA-1 (pre-Alpha wild-type), Omicron and number out of total that neutralized Omicron was abstracted from studies. Figure 4 illustrates the need of harmonizing the neutralization titer readout other than using the mean of each study. Relative titers as suggested above may be a solution. An amended Figure 4 should also address dynamic ranges of the observed titer and comment on within study variability.

We do agree that a relative titer with minimum and maximum is an alternative. We kept the original figure 4 and added new figures including the minimum and maximum along with the GM(GMT50).

5. The opportunity to go beyond a simple neutralization titer assessment was missed. Considering all the data the authors extracted from the published studies, a meta-analysis to assess factors associated with neutralizing titers against omicron / wt could be considered and may provide interesting insights.

We fully agree that interesting insights could stem by correlation between clinical variables and nAb titer, but we can unfortunately confirm that none of the *in vitro* studies we reviewed attempted such associations. We are so far limited at the time between infection/vaccination and serum sampling.

6. The analysis is kept mostly descriptive, more quantitative measures would be important to incorporate. In fact, the interpretation of the data given in the results is not on all occasions the one an independent reader may see when looking at the figures ...

We have added quantitation where possible along with live virus or pseudovirus in tables. We are now providing as a Supplementary appendix an Excel file with our source data for all figures and tables. We believe tables and figures report a full set of quantitative measures such as GMT, fold reductions and number of plasma units which neutralize omicron as well as minimum and maximum values in the assays.

a. Line 29: "The novel SARS-CoV-2 Omicron, with its antigenic escape from unboosted vaccines and therapeutic monoclonal antibodies, demonstrates the continued relevance of COVID19 convalescent plasma (CCP) therapies" Given the current information on clinical use

of CCP it is very bold to state “demonstrates the continued relevance of COVID19 convalescent plasma (CCP) therapies”. This needs to be reworded.

We agree that clinical use of CCP is (unfortunately) still spotty, but we are here referring to the therapeutic potential of CCP, not its current adoption into clinical practice. While it is now clear that immune escape has caused failures of most mAbs, VaxCCP has preserved efficacy. So, we feel our wording is logical when referring to post vaccination/post COVID-19 plasma. We hope that publication of this study will solicit the initiation of novel randomized controlled trials.

b. “Line 43 Thus, CCP provides an effective tool to combat the emergence of variants that defeat therapeutic monoclonal antibodies”. Why/how would CCP combat the emergence of variants? This statement needs explanation and is not suited for the abstract.

Yes, we have deleted “the emergence”, since the intended meaning was CCP utility as a therapeutic agent. In this sense, we feel it is suited for the abstract.

c. “Line 96 Consequently, a more in-depth analysis is needed to better stratify the populations.” This statement is unclear and suggests that the present study provides a means of clinically relevant stratification of CCP, which is not the case.

Changed “populations” into “CCP types” for better clarity.

8. Tables 2 through 9 are not mentioned in the results!

As stated in the cover letter, at the time of initial submission we had left the editor with the opportunity to consider Tables 2-9 as Supplementary Tables. We have now decided to proceed this way, so they are cited in the Results section as Supplementary Tables 1 to 8. We added tables on quantitative measures for BA.1, BA.2 and BA.4/5 neutralization numbering up to 12 tables describing each study with extracted data.

9. Line 113 “...with a few exceptions”. Please clarify: which studies used replication competent virus?

We have added references at studies which used live authentic SARS-CoV-2 in neutralization assays and added a column in the supplementary tables to discriminate studies relying over authentic live virus versus pseudovirus neutralization assays.

10. Line 118 “...nAb titers of 850 to 2,000” Please clarify: NT50? GMT? Nomenclature keeps changing in the result section. Please consider relative titers for comparability.

We have changed every recurrence of GMT into GMT₅₀ (geometric mean of 50% neutralization titres) to better clarify the method used in the original research papers.

11. Line 119 “The same plasma”Please clarify: Does this refer to the Beta VoC infected CCP?

We have clarified that it referred to both any type of pre-Omicron CCP and 2-dose vaccinee plasma.

12. Line 154 onwards: “Few data exist for comparisons among different vaccine boosts. For CoronaVac® (SinoVac), three doses led to 5.1 FR in nAb titer 20, while for Sputnik V nAb titer moved from a 12-fold reduction at 6-12 months up to a 7-fold reduction at 2-3 months after a boost with Sputnik Light.” Please clarify “5.1 FR in nAb titer 20”, what does this refer to exactly?

“20” was a reference number. We have added the comparator (wild-type) for fold-reduction for better clarity.

13. Line168: “IC50 values to mean 1:2929 at around 9-12 days, which were higher than the mean peak IC50 values of BNT162b2 vaccines” Referring here to IC50 makes these data difficult to other parts in the result section. As mentioned above, working with relative titers would resolve this issue.

Apologies, this should have been a GMT₅₀, as usual. This data is now within Table 2.

14. Line 34 approximately 50% (426/841). Please clarify: are these treated patients?

As in the entire manuscript, we are referring here to *in vitro* neutralization, as a surrogate for treated patients.

15. Line 35 “...about 30”. Please clarify: Does this mean 30 NT50? Or GMT?

Clarified the unit of measurement was GMT₅₀.

16. Line 35/36 Two doses of mRNA vaccines had a similar 50% percent neutralization with more than doubling of Omicron neutralization mean titer (about 60) Please clarify: This refers to neutralizing activity of CCP of vaccinated patients? How long after vaccination? (Latter needs to be also stated in results).

Better clarified that this refers to 2-dose vaccine recipients that were not convalescent. As shown in Supplementary Tables 5 and 6, time since last dose (when disclosed by the authors of the references studies) ranged from 1 to 5 months. We have now extracted the data from studies reporting it.

17. Line 121: “The nAb FR against Omicron was now 10 to 20 ...” What do the authors trying to say?

Changed “now” into “in the latter subgroup” for better clarity

18. Line 213 “...likely (less than 50%)”. Please clarify, less than 50% or likely not?

Corrected to “unlikely”.

19. Figure legend 2. Panel A and B are not explained.

Figure legend for panels A and B now explained (“A) unvaccinated convalescent plasma and B) vaccinated plasma with or without COVID-19.”)

20. Figure legend 4. Description of panel C is missing

Figure legend for panel C now added (“C) 9 additional studies looked at the same vaccine conditions in the first 5 comparing WA-1 nAb to Omicron nAb.”)

21. Table 4-10: Please clarify in Table what “mean” stands for. Geometric mean neutralizing titer (GMT)? What does time stand for? Sampling time after last vaccination dose? Fold drop means fold reduction to WA-1? This is not clearly labelled in all tables. Please use the same terminology in text and tables.

Table headings have been expanded for better clarity and double-checked for consistency. “Time” was the time between last vaccination or COVID-19 and serum sampling: we hence clarified the table header into “blood sample interval after vaccine or COVID-19” where available.

REVIEWERS' COMMENTS

Reviewer #1 (Remarks to the Author):

Thank you for providing detailed answers and clarifying the text.

I have some minor points:

In your abstract please minimise the use of acronyms (GMT) and please explain what Vax-CCP (VaxCCP in last line) is.

Vax-CCP or VaxCCP? Please define and keep throughout. Please also make sure that you use the right term - for example about using CCP in LMICs should be Vax-CCP, correct?

Line 231 "(" should be ("

Please include the information about normality of the datasets in your statistics section within methods.

The new Figure 1 is much clearer thanks.

Please add the clarification about the missing data from Lusvarghi in the legend.

Line 617: IgA and IgM not present, please clarify the relevance. If this is concentrated IgG there will not be IgA or IgM. It is later mentioned 'which have powerful SARS-CoV-2 activity⁴⁸⁻⁴⁹' – I think that belongs here? Please modify to 'Vax-CCP' rather than 'VaxC-CP'.

Line 628 'pseudovirus'

Figure 6 is quite small and difficult to read.